# The Effect of EEG Biofeedback Training Frequency and Environmental Conditions on Simple and Complex Reaction Times

**DOI:** 10.3390/bioengineering11121208

**Published:** 2024-11-29

**Authors:** Skalski Dariusz, Maciej Kostrzewa, Prończuk Magdalena, Jarosław Markowski, Jan Pilch, Marcin Żak, Adam Maszczyk

**Affiliations:** 1Institute of Pedagogy and Health Sciences, University of Applied Sciences in Wałcz, Wojska Polskiego 99, 78-600 Wałcz, Poland; dariusz.skalski@pwsz.eu; 2Department of Physical Culture, Gdansk University of Physical Education and Sport, Kazimierza Górskiego 1, 80-336 Gdańsk, Poland; magdalena.pronczuk@awf.gda.pl; 3Institute of Sport Sciences, The Jerzy Kukuczka Academy of Physical Education, Mikolowska 72a, 40-065 Katowice, Poland; m.zak@awf.katowice.pl; 4Department of Laryngology, Faculty of Medical Sciences, Medical University of Silesia in Katowice, Medyków 18, 40-752 Katowice, Poland; jmarkowski@sum.edu.pl; 5Department of Physiological and Medical Sciences, The Jerzy Kukuczka Academy of Physical Education, Mikolowska 72a, 40-065 Katowice, Poland; j.pilch@awf.katowice.pl

**Keywords:** normoxia, normobaric hypoxia, neurofeedback training

## Abstract

The objective of this study is to evaluate the impact of EEG biofeedback training under normoxic and normobaric hypoxic conditions on both simple and complex reaction times in judo athletes, and to identify the optimal training frequency and environmental conditions that substantially enhance reaction times in the examined athlete groups. The study comprised 20 male judo athlete members of the Polish national judo team in the middleweight and heavyweight categories. We randomly assigned participants to an experimental group and a control group. We conducted the research over four cycles, varying the frequency of EEG biofeedback sessions and environmental circumstances for both the experimental and control groups. Every research cycle had 15 training sessions. The results showed that the experimental group, following the theta/beta regimen, got significantly faster at complex reactions after a training cycle that included sessions every other day at normal oxygen levels. Following daily training sessions in normoxic circumstances, we noted enhancements in simple reaction speeds. Under normobaric hypoxia conditions, the judo athletes showed deterioration in both simple and complex reaction times. The control group showed no similar changes. Daily EEG training in normoxic settings markedly improved simple reaction time, but EEG-BF training conducted every other day greatly raised complicated reaction time. In contrast, training under normobaric hypoxia settings did not result in enhancements in basic or complicated reaction times following EEG training.

## 1. Introduction

High concentration and the associated heightened alertness in athletes, achieved under optimal arousal of their nervous system, enhance cognitive processes and contribute to shorter reaction times [1,2,3,4]. EEG biofeedback (EEG-BF), also referred to as neurofeedback, is a technique that enables individuals to gain control over specific brainwave frequencies through real-time feedback. This method relies on operant conditioning, where positive reinforcement is provided when desired brainwave patterns are achieved, facilitating changes in neural activity and enhancing cognitive functions [5]. Numerous studies report improvements in reaction time following EEG biofeedback training; however, the applied research protocols vary significantly [5,6,7,8].

Scientific reports have demonstrated positive changes in the effects of hypoxia, including improvements in blood buffering capacity, increased activity of oxidative and glycolytic enzymes, an enhanced capillarization of muscle fibers, improved glycolytic capacity, and improved myoglobin content in muscles [9]. As a result, normobaric hypoxia conditions have been utilized for rehabilitation purposes, such as reducing complications following ischemic stroke [10], as well as for decreasing risk factors of cerebrovascular diseases [11]. Some studies suggest that training in a normobaric hypoxic environment can optimize athletes’ functional state and enhance performance during competitions [12]. Additionally, immediately following hypoxic exposure, a reperfusion effect may occur, which could improve cognitive function, as described in both animal models [13] and humans [11,14]. However, research on the effects of normobaric hypoxia on cognitive function remains inconclusive. Nevertheless, growing evidence suggests that moderate hypoxia may not have a detrimental effect on cognitive abilities [15,16]. It is essential to consider the previously described high individual variability in response to hypoxia exposure [17]. Moreover, the response to normobaric hypoxia exposure appears to be largely dependent on the type and protocol of hypoxia (hypobaric/normobaric, intermittent/continuous), participant age, fitness level, health status, type of cognitive task, timing of the post-exposure test, and other confounding factors [18,19]. Therefore, adjusting the appropriate timing and frequency of training is crucial to ensure a positive response to hypoxic exposure [20]. This principle also applies to training under normoxic conditions, where the frequency and duration of sessions are of fundamental importance [21].

An analysis of the existing literature evaluating the effectiveness of EEG-BF training in improving reaction times in elite athletes reveals a limited number of studies and training protocols. Available scientific reports investigating the impact of EEG-BF training on increasing beta wave amplitude have been applied to improve cognitive functions in older adults or enhance attention retention in adults. There is also research indicating that greater beta wave amplitude (especially in the parietal regions) prior to visual stimulus exposure was associated with shorter reaction times [22]. However, these studies were not aimed at improving reaction times in elite athletes. Considering the conceptual framework provided by Wróbel [23] and the lack of studies on the effectiveness of EEG-BF in enhancing beta band amplitude to improve reaction times in elite judo athletes, the present study aims to fill this gap. The mechanism of EEG-BF involves modulating specific brainwave frequencies to optimize brain function. For example, increasing the amplitude of beta waves (13–30 Hz) has been linked to improved focus, alertness, and faster reaction times, while reducing theta waves (4–7 Hz) has been associated with decreased mind wandering and enhanced concentration [23].

Given the limited research on the effectiveness of EEG-BF training in elite athletes, particularly in improving reaction times, this study fills an important gap in the literature. Building on the work of Wróbel [23] and others, this research aims to establish the efficacy of EEG-BF in optimizing cognitive and motor performance in judo athletes under normoxic and normobaric hypoxic conditions. By identifying the optimal training protocols and environmental factors, the findings may contribute to the development of tailored programs that enhance competitive performance.

This study sought to examine the effects of EEG biofeedback training on the simple and complex reaction times of judo athletes in normoxic and normobaric hypoxic conditions. The aim was to determine the optimal training frequency and environmental factors that enhance reaction times in the examined athlete groups. This study examines the efficacy of EEG biofeedback in enhancing cognitive and motor performance, providing significant insights for coaches and sports scientists involved with judo athletes. The findings may enable the development of tailored training programs that improve athletic performance in competitions.

## 2. Materials and Methods

### 2.1. Participants

The study sample consisted of 20 male judo practitioners, members of the Polish national judo team in the middleweight and heavyweight categories. Inclusion criteria mandated a minimum of six years of training experience and a six-month hiatus from altitude training prior to participation. All participants were medically certified to be in good health and fit for high-intensity exercise. The study took place during two consecutive preparatory phases following the primary competitions of the 2023 and 2024 seasons. Measurements were consistently conducted in the morning between 10:00 and 14:00, with individual assessments for each athlete. Participants were instructed to abstain from caffeine and stimulants, including energy drinks, throughout the training cycles.

Subjects were randomly allocated to an experimental group (EG) and a control group (CG) (Table 1). Before commencing, participants were fully informed of the study’s aims and procedures, provided signed consent for participation, and were made aware of their right to withdraw from the study at any time without providing justification.

This research was conducted under grants N RSA3 03953 and N RSA4 04054, with ethical approval from the Bioethics Committee for Scientific Research at the Academy of Physical Education in Katowice. Experimental procedures were carried out in the Human Psychomotor Performance Laboratory and the Hypoxia Laboratory at the Jerzy Kukuczka Academy of Physical Education in Katowice.

### 2.2. Research Protocols

The research was organized into four separate cycles. Each cycle changed the number of EEG biofeedback sessions and the environmental conditions (normoxia vs. normobaric hypoxia) for both the experimental and control groups.

During the initial cycle, we conducted EEG biofeedback training sessions every two days under normoxic settings. The second cycle maintained this frequency, albeit with sessions conducted under normobaric hypoxia. The third cycle comprised daily EEG biofeedback sessions in normoxic conditions, whereas the fourth cycle contained daily training under hypoxic conditions. The hypoxic training happened in a controlled setting with a normobaric hypoxia-generating system (LOS-HYP1/3NU, Lowoxygen Systems, Berlin, Germany) that made it feel like the participants were 2500 m above sea level (FiO_2_ = 15.5%), which is a level that does not affect brain function [24]. Each cycle consisted of 15 training sessions, followed by a four-week hiatus, adhering to a modified Thompson training protocol [25]. Training sessions performed under normobaric hypoxia lasted 20 min, consistent with the advised period that preserves cognitive function [26].

The principal training regimen for the experimental group was theta/beta training, designed to improve concentration and facilitate a “narrow focus” in athletes. The control group followed the identical schedule, session length, and frequency of EEG biofeedback sessions as the experimental group. Instead of participating in the theta/beta technique, the control group observed a simulated EEG, unrelated to the individuals’ actual brain wave patterns. We performed all control group sessions solely under normoxic circumstances.

### 2.3. EEG Recording

Biofeedback training was conducted with EEG DigiTrack software (ver. 15.1) in conjunction with an ExG-32 head. ISO and CE medical certifications have verified the equipment’s quality. Prior to each EEG signal recording, scientists assessed electrode impedance and interelectrode impedance using an integrated impedance sensor. To initiate EEG biofeedback diagnoses and training, it is essential to attain an impedance level below 5 KΩ and an interelectrode measurement variance that is not above 1 KΩ. Every training session for the designated cycle began with a 5 min single-channel assessment. During this period, investigators instructed the participant to maintain an open gaze for one minute, thereafter closing their eyes for another minute, and then reopen their gaze while performing an activation task of counting backwards by seven from one hundred. The diagnostic technique positioned the reference electrode on the left earlobe, the ground electrode on the right earlobe, and the active electrode at point Cz, following the international 10–20 system. We acquired electroencephalogram (EEG) data from the C3 location on the scalp using Ag/AgCl electrodes (Blue Sensor SP, Ambu, Ballerup, Denmark). Receiving the signal from C3 made it easier to reach the training goal of finding the best balance between the speed of rapid waves (beta) and slow waves (theta) in the prefrontal cortex. These waves govern attention and concentration. The active electrodes utilizing a stretchy lycra cap were positioned, conforming to the 10–20 system. Everyone had conductive gel administered under the electrodes using a blunt needle, and then the electrode locations were disinfected using an abrasive cream (Nuprep, Weaver and Company, Aurora, CO, USA) and alcohol wipes.

### 2.4. EEG Biofeedback Training

Participants completed the Mood and Six Emotions Questionnaire prior to each training session, and a neurologist interpreted the data to assess training eligibility. Each session began with reaction time assessments. Participants subsequently engaged in EEG biofeedback training under either normoxic or normobaric hypoxic settings. Each session involved positioning participants 70 cm away from a 17-inch LCD monitor that displayed the EEG biofeedback animation. The monitor constantly monitored electrode impedance and directed participants to refrain from head movements. The session commenced following the placement of EEG electrodes, followed by a 2 min rest interval to acclimate participants to the training environment and to capture a non-training EEG sample.

The EEG-BF training involved recording neural activity from nerve cells as electrical impulses, which the computer software then processed into amplitude values within specific frequency ranges. All athletes executed the identical task during the EEG-BF training, maneuvering an on-screen vehicle down a roadway. The EEG-BF training initiated the vehicle’s motion when it detected the requisite EEG signal amplitude from two electrodes, C3 and C4 (as per the 10–20 system), inside the beta and theta frequency regions. Upon the alteration of the wave amplitude in the designated direction, the vehicle advanced. The threshold delineating the requisite beta and theta wave amplitude was illustrated as a horizontal line on the graph, ensuring that the threshold remained above the upper limit of the beta band and below the lower limit of the theta band, thereby providing a relatively stable level of reward for all participants. The athlete concentrated on the task shown on the monitor and received feedback regarding the bioelectric activity from the recorded brain region. Upon attaining the ideal brainwave frequency pattern, participants received points, indicated by an auditory cue, but points were not provided when undesirable frequency bands prevailed. The brain’s bioelectric signals were analyzed to establish feedback between visual and auditory stimuli and the trainee’s bioelectric brain response. This feedback resulted in modifications to brain bioelectric activity (neuronal activity) to achieve the required level. Following the EEG biofeedback training, reaction time assessments were conducted utilizing specific trials from the Vienna Test System (VTS).

### 2.5. Reaction Time Tests

The impact of EEG biofeedback training on the reaction times of judo athletes was evaluated using specific trials from the Vienna Test System (VTS). Assessments were performed immediately prior to and following the EEG biofeedback training. Each test was conducted twice, with a 5 min gap, and the superior result from the two measures was taken into account. The Vienna Test System employed a reaction time device (RT) to assess simple visual reaction times. The participant was required to swiftly transition their hand from the “rest key” to the “reaction key” upon the illumination of a yellow light. The average reaction time was computed in seconds using the gathered data. Complex reaction times were assessed using the Decision Reaction Test (DG), in which participants were required to promptly push the key corresponding to the color of the light upon stimulus presentation. The program documented all accurate and erroneous responses, the mean reaction time, and the standard deviation of the mean reaction time. The stimulus was presented 15 times.

### 2.6. Statistical Methods

Basic descriptive statistics were computed to characterize the structure of the examined variables, encompassing measures of central tendency—arithmetic mean (x)—and measures of variability—standard deviation (S). The results and input data are shown in a tabular matrix format.

The distribution of the examined variables was assessed utilizing the Shapiro–Wilk normality test. Levene’s test was employed to assess the homogeneity of variance. In summary, all variable variances exhibited normal distributions with minor left or right skewness, remaining within the normal range. The significance level for Mauchley’s test of sphericity was evaluated, and due to the results being statistically insignificant, sphericity of variance was presumed.

In the first stage of the empirical analysis, to examine the dynamics of the phenomenon, relative single-base growth rates were used, in which the values of the analyzed variables were considered as a function of time, and the effect size of the training process was determined using the Eta-squared (ɳ^2^) measure. By assigning a numbering system from t = 0 to t = n − 1 to the time units and the observed levels of the studied phenomenon, a so-called realization of the stochastic process over time was obtained, and the effect size was determined according to the assumption for ɳ^2^, where 0.01 represents a small effect, 0.06 represents a medium effect, and 0.14 represents a large effect.

To verify the significance of differences between the groups, a repeated measures analysis of variance (ANOVA) was used. If significant differences were found, a post hoc Tukey test was performed for a further analysis. The F-statistic and the significance level are presented. For all analyses, a significance level of *p* < 0.05 was adopted. All calculations were carried out using Statistica v.13 software (StatSoft, 2021, Tulsa, OK, USA).

## 3. Results

The comparative characteristics of changes in simple and complex reaction times due to EEG biofeedback training procedures in normoxic and hypoxic conditions among judo athletes, along with a within-group variance analysis, are detailed in Table 2 and Table 3 and Figure 1, Figure 2, Figure 3 and Figure 4.

## 4. Discussion

In recent decades, several studies have been undertaken to devise new strategies that enhance the training process aimed at improving reaction speeds to visual and aural stimuli [3]. A meticulous selection of training parameters aligned with the athlete’s capabilities is vital for the balanced enhancement of fitness, preservation of health, and cultivation of motor skills, which are fundamental for future athletic success [27]. Athletes, having undergone extensive technical, tactical, physical, and psychological preparation, are pursuing innovative ways to gain a competitive advantage over their adversaries. A potential option is the implementation of EEG biofeedback (EEG-BF) training in normoxic and normobaric hypoxic environments.

The point of this study was to find out how training with EEG-BF in normoxic and normobaric hypoxic environments affected judo athletes’ ability to improve their simple and complex reaction times. This dissertation’s original feature is the implementation of EEG-BF training under both normoxic and normobaric hypoxic settings. We performed the study on a specific cohort of athletes from the Polish Judo Association national squad, all of whom hold international master-level titles.

Due to the significant role that reaction speed plays in judo athletes’ effectiveness during combat, this study aimed to examine the influence of EEG-BF training, focused on enhancing fast beta waves and inhibiting slow theta waves, on their reaction times. Compelling evidence suggests that low theta wave amplitudes correlate with improved memory, which may positively impact cognitive function [28]. Based on previous analyses, which showed that higher beta wave activation (especially in the parietal areas) before the exposure to a visual stimulus was associated with shorter reaction times [22], we conducted an experiment using a protocol focused on enhancing beta1 wave amplitude and lowering theta wave amplitude to improve both simple and complex reaction times.

The results indicated that the experimental group, following the theta/beta protocol, achieved significant improvements in complex reaction times after a training cycle with sessions held every other day in normoxic conditions. Improvements in simple reaction times were observed after daily training sessions under normoxic conditions. Under normobaric hypoxic conditions, the judo athletes showed worsened simple and complex reaction times. No similar changes were observed in the control group.

The analyses revealed the greatest improvement in reaction times under normoxic conditions after 15 training sessions. Similar observations were made by Rostami et al. [29], where 15 EEG-BF training sessions using the SMR and alpha/theta protocols, combined with beta2 wave suppression, significantly improved shooting accuracy in a group of rifle shooters compared to a control group [29].

In a study by Vernon et al. [30], fewer sessions were used—only eight. These studies confirmed previous findings suggesting beneficial changes in modulating SMR wave amplitude after just eight training sessions, without changes in theta wave amplitude [30]. This is consistent with the results of Reis et al. [31], which demonstrated that a short, intensive EEG-BF training protocol consisting of eight 30 min sessions significantly modulated alpha and theta wave amplitudes, improving cognitive functions in the tested group [31].

Jurewicz et al. [32] used 16 training sessions but observed the greatest increase in beta1 frequency wave amplitude after the 3rd EEG-BF session, after which the increase plateaued [32].

Scientific reports also suggest that changes can occur after a single EEG-BF session [33]. Mottola et al. [34] confirmed this, showing that a single 12 min biofeedback session significantly improved endurance exercise performance, with the experimental group cycling 30% longer on an ergometer than the control group [34].

Discrepancies in the length, frequency, and number of training sessions required to achieve training goals largely stem from individual differences, so the optimal approach is to develop personalized EEG-BF interventions for each athlete.

Gruzelier et al. [5] analyzed the impact of beta1 wave stimulation at point C3, showing positive changes in the form of fewer errors and less variability in reaction times [5]. Later studies confirmed that increased beta wave amplitudes (especially in the parietal areas) before exposure to a visual stimulus were associated with shorter reaction times [22]. These findings align with the results of the present study, which demonstrated the positive effects of enhancing beta1 wave activity at point C3 on improving simple and complex reaction times in judo athletes.

Our analyses showed the greatest improvement in complex reaction times after the first training cycle, with sessions held every other day. The shortest simple reaction times were observed after daily training sessions. These results can be explained by the task’s complexity—simple tasks, which involve a single, standard response to a specific signal, are easier and improve faster with EEG-BF training. Simple reactions do not require as many neural pathways as complex reactions, making it easier to achieve sufficient synaptic transmission efficiency to improve response times to single stimuli [35].

Applying a training configuration with increased session frequency and fewer rest periods seems optimal for shortening simple reaction times. To achieve similar results in complex reaction times, which involve multiple stimuli and more advanced processing, it is necessary to engage and expand more complex neural networks [36]. Such effects typically occur after a longer training period, with sufficient rest breaks to allow the restructuring and consolidation of beneficial changes in synaptic connections.

In our study, we observed longer simple and complex reaction times after exposure to normobaric hypoxia. Similar findings, where hypoxia negatively affected reaction times, were reported by Dart et al. [37] and Dykiert et al. [38]. Conversely, Chroboczek et al. [39] and Taylor et al. [26] found no differences in test performance under hypoxic conditions. Differences in research results regarding the effects of normobaric hypoxia on cognitive abilities and reaction times may stem from various research protocols, exercise selections, and hypoxia intensities.

Moreover, in our study, cognitive tests were performed immediately after hypoxic exposure, where a reperfusion effect and subsequent improvement in cognitive function were expected, as described in humans [11,14]. The mechanism of cognitive improvement after exposure may be due to increased oxygen, energy, and/or hormone supply to neural tissue resulting from increased blood flow [40]. Furthermore, moderate oxidative stress and inflammation induced by hypoxic exposure can modulate the synthesis and release of BDNF and prevent hippocampal impairment [20]. Hypoxia-induced BDNF production may facilitate memory functioning by improving synaptic strength and, consequently, enhancing cognitive functions.

It seems that the response to normobaric hypoxia exposure largely depends on the mode and protocol of hypoxia (hypobaric/normobaric, intermittent/continuous), participant age, fitness level, health status, cognitive task type, test timing, and other confounding factors [18,19,41,42,43]. There remain many areas of research that need to be explored to fully understand the impact of EEG-BF training on athletes’ training effectiveness. Therefore, new research directions should be identified to develop optimal EEG-BF training protocols in normobaric hypoxic conditions, which could maximize athletes’ performance.

## 5. Conclusions

In the judo athlete groups under study, the EEG-BF training in normoxic settings significantly influenced both simple and complex reaction times.

Daily EEG-BF training under normoxic settings markedly improved simple reaction time, but EEG-BF training conducted every other day greatly raised complicated reaction time. In contrast, training in normobaric hypoxia did not enhance simple or complicated reaction speeds following EEG-BF training.

## 6. Study Limitations

The small sample size is a limitation of this study, although it was dictated by the selection of elite international-level athletes. Future studies could include athletes with lower training levels to observe the effects of neurofeedback training in various configurations. Additionally, including a control group that receives no intervention, in addition to the sham group, would help avoid assumptions of a placebo effect when assessing the training cycle’s effectiveness.

Further research should focus on extending the training period and considering the long-term dynamics of reaction time variability, which could further substantiate the findings of this study.

Another limitation was the absence of measurements for cerebral blood flow (CBF) and brain-derived neurotrophic factor (BDNF). Hypoxia can modulate BDNF synthesis and release, preventing hippocampal impairment [20]. Hypoxia-induced BDNF production may enhance memory function by improving synaptic strength, consequently enhancing cognitive functions.

There remain many research directions that need to be pursued to fully understand the impact of EEG-BF training on athletic performance. Exploring these new directions will help develop optimal EEG-BF training protocols under normobaric hypoxic conditions, potentially maximizing athletes’ sports performance.

## Figures and Tables

**Figure 1 bioengineering-11-01208-f001:**
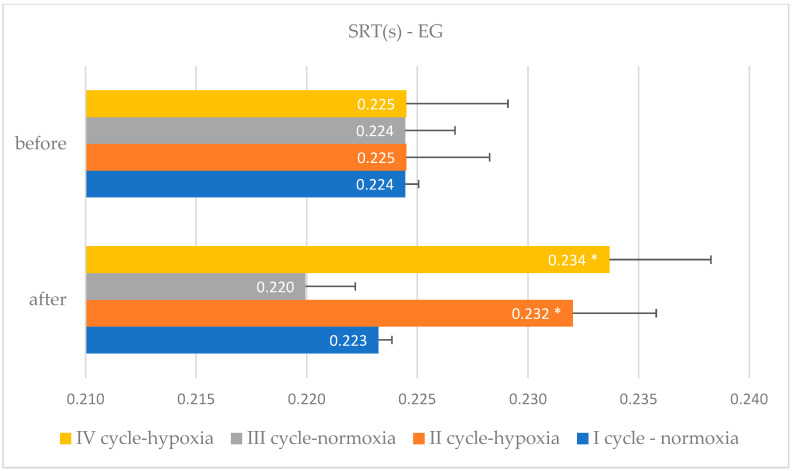
Simple reaction time changes in the experimental group after EEG biofeedback training in normoxic and hypoxic conditions—cycle I–IV; *—significant changes.

**Figure 2 bioengineering-11-01208-f002:**
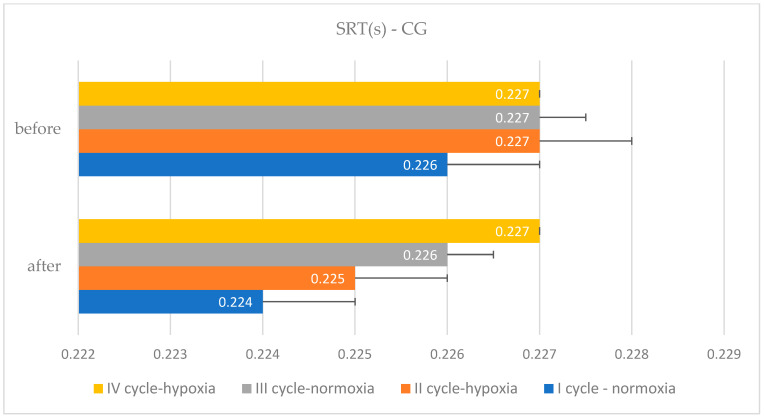
Simple reaction time changes in the control group after Eeg biofeedback training in normoxic and hypoxic conditions—cycle I–IV.

**Figure 3 bioengineering-11-01208-f003:**
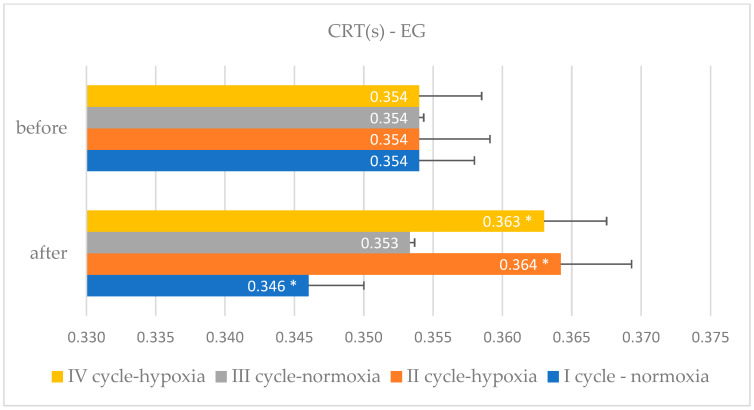
Complex reaction time changes in the experimental group after EEG biofeedback training in normoxic and hypoxic conditions—cycle I–IV; *—significant changes.

**Figure 4 bioengineering-11-01208-f004:**
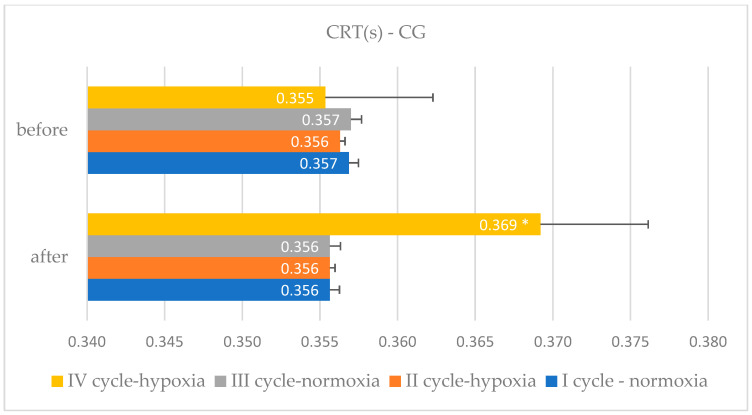
Complex reaction time changes in the experimental group after EEG biofeedback training in normoxic and hypoxic conditions—cycle I–IV; *—significant changes.

**Table 1 bioengineering-11-01208-t001:** Characteristics of Experimental and Control Groups.

Group	n	Age (y)	Height (cm)	Weight (kg)	FAT (%)
EG	10	19.6 ± 1.4	182.2 ± 5.1	78.6 ± 6.9	10.1 ± 3.7
CG	10	19.8 ± 1.2	181.8 ± 4.5	77.1 ± 6.1	9.8 ± 4.1

**Table 2 bioengineering-11-01208-t002:** Characteristics of Variations in Simple Reaction Time (s) Pre- and Post-EEG Biofeedback Training in Normoxic and Hypoxic Conditions Among Judo Athletes Across Distinct Training Cycles.

Simple Reaction Time (SRT)					
Group	Cycle	Pre (s)	Post (s)	*p*	ɳ^2^
EG	I	0.224	0.223	0.821	0.080
	II	0.225	0.232	0.001	0.160
	III	0.224	0.219	0.001	0.140
	IV	0.224	0.234	0.001	0.181
CG	I	0.226	0.224	0.721	0.040
	II	0.227	0.225	0.703	0.041
	III	0.227	0.226	0.721	0.031
	IV	0.227	0.227	0.911	0.010

EG: Experimental group; CG: Control group; SRT: Simple reaction time; *p*-value: Statistical significance; ɳ^2^: Eta squared, a measure of effect size.

**Table 3 bioengineering-11-01208-t003:** Characteristics of Variations in Complex Reaction Time (s) Pre- and Post-EEG Biofeedback Training in Normoxic and Hypoxic Conditions Among Judo Athletes Across Distinct Training Cycles.

Complex Reaction Time (CRT)					
Group	Cycle	Pre (s)	Post (s)	*p*	ɳ^2^
EG	I	0.354	0.346	0.017	0.14
	II	0.354	0.364	0.001	0.18
	III	0.354	0.353	0.317	0.02
	IV	0.354	0.363	0.001	0.17
CG	I	0.357	0.356	0.911	0.02
	II	0.356	0.356	0.999	0.01
	III	0.357	0.355	0.887	0.02
	IV	0.355	0.369	0.024	0.08

ɳ^2^: a measure of effect size.

## Data Availability

All data are located at the premises of AWF Katowice, 40-065 Katowice; Mikołowska 72a, and can be provided upon specific request from researchers.

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
