# Peer review of "The Effect of EEG Biofeedback Training Frequency and Environmental Conditions on Simple and Complex Reaction Times"

_bioengineering, 2024, doi:10.3390/bioengineering11121208_

Round 1

Reviewer 1 Report

Comments and Suggestions for Authors

I appreciate the authors for presenting this valuable research article, which focuses on assessing the effects of EEG biofeedback training on simple and complex reaction times in judo athletes under normoxic and normobaric hypoxic conditions.  The results suggest that regular, personalized biofeedback training in normoxic conditions may enhance athletic performance in judo athletes.

 My comments are as follows:

 1. A primary limitation of the current study is (a) the small sample size, with only 20 athletes (10 per group). This may not be sufficient to account for individual variability. (b) factors such as sleep quality, stress, or dietary differences could still influence reaction times and EEG outcomes. (3) Additionally, the study did not explore other hypoxic intensities conditions, which might yield insights into how different types of hypoxia impact cognitive performance.

2. The differences in reaction times fall within a narrow range of 0.01 to 0.001 seconds (e.g., 0.354 vs. 0.346 or 0.224 vs. 0.234 seconds). Although these differences are statistically significant, further clarification on their clinical significance is warranted. This is my major concern

Author Response

A primary limitation of the current study is (a) the small sample size, with only 20 athletes (10 per group). This may not be sufficient to account for individual variability.

1 Small sample size

We acknowledge that the small sample size is a limitation of our study. Our research was conducted on members of the Polish national judo team in the middleweight and heavyweight categories. Due to the specific and elite nature of this group, the maximum number of available athletes is limited to 18 across both weight categories. Nevertheless, we sought to mitigate this limitation by employing rigorous inclusion criteria and matching the groups in terms of athletic level and experience.

(b) factors such as sleep quality, stress, or dietary differences could still influence reaction times and EEG outcomes.

2 Influence of additional factors (sleep, stress, diet)

We appreciate this point and agree that factors such as sleep quality, stress levels, and diet could influence reaction times and EEG outcomes. During the study, we attempted to monitor these variables using questionnaires on sleep quality and subjective fatigue levels. However, detailed analyses of these factors were beyond the scope of the current study and represent an avenue for future research.

(3) Additionally, the study did not explore other hypoxic intensities conditions, which might yield insights into how different types of hypoxia impact cognitive performance.

3 Lack of analysis of other hypoxic intensities

We agree that different levels of hypoxia could have varying effects on cognitive abilities and reaction times. In this study, we chose to standardize the hypoxic conditions (simulating an altitude of ~2,000 meters above sea level) to ensure comparability of results. We are considering expanding future research to include various levels of hypoxia.

The differences in reaction times fall within a narrow range of 0.01 to 0.001 seconds (e.g., 0.354 vs. 0.346 or 0.224 vs. 0.234 seconds). Although these differences are statistically significant, further clarification on their clinical significance is warranted. This is my major concern

4 Clinical significance of reaction time differences

The reaction time differences (e.g., 0.354 vs. 0.346 s) indeed fall within a narrow range. However, in the context of elite sports, particularly in disciplines like judo, fractions of a second can determine success or failure—for example, in executing techniques or evading an opponent’s attack. Although these differences may appear small, they are statistically significant and potentially meaningful from a practical sports performance perspective. We have elaborated on this aspect in the “Discussion” section of our manuscript to better justify the relevance of our findings in the context of elite sports.

Once again, we thank you for your valuable feedback. We hope the revisions and explanations provided address your concerns regarding our study.

Reviewer 2 Report

Comments and Suggestions for Authors

This study aims to investigate the impact of EEG biofeedback (EEG-BF) training on reaction times in judo athletes. The study examines how EEG-BF training conducted under normoxic and normobaric hypoxic conditions affects both simple and complex reaction times. The methods employed are based on a precise research design that monitored the cognitive functions of athletes throughout training sessions and compared the results.

The findings revealed that EEG-BF training under normoxic conditions significantly improved both simple and complex reaction times. Conversely, training under normobaric hypoxic conditions did not contribute to reaction time improvements. These results demonstrate how the frequency and environmental conditions of EEG-BF training influence reaction times, providing valuable insights for enhancing athletes’ training strategies. The study suggests that proper EEG-BF training can enhance reaction times, potentially improving athletic performance, making these findings highly intriguing.

This research addresses an important topic concerning the interaction between athletic training and cognitive functions, complementing existing studies. Therefore, the paper holds significant social value, as it contributes to the potential improvement of training methods for athletes.

However, there are a few areas that require revision:

Firstly, regarding the results section, the comparative illustrations for each group (Figure 1-4) should be shown not only as averages but also using box plots to display all sample data or by including statistical details such as error bars. This would allow readers to assess the appropriateness of the measurements, sample size adequacy, and any potential biases, especially given the notable differences between significant and non-significant p-values.

Secondly, the introduction lacks detailed explanations and citations regarding EEG-BF training. Adding more information and appropriate references about the basics and background of EEG-BF training would make the introduction more comprehensive and reader-friendly.

By addressing these revisions, this study will become an even more valuable report for readers. After that, I recommend this study for publication.

Author Response

Firstly, regarding the results section, the comparative illustrations for each group (Figures 1-4) should be shown not only as averages but also using box plots to display all sample data or by including statistical details such as error bars. This would allow readers to assess the appropriateness of the measurements, sample size adequacy, and any potential biases, especially given the notable differences between significant and non-significant p-values.

Thank you for your suggestion regarding data visualization in the results section. We agree that presenting the data not only as means but also using box plots would allow readers to better understand the distribution of results, data variability, and potential measurement errors. We have implemented the recommended changes by adding box plots for each group and supplementing them with statistical details such as error bars and value ranges. We hope this approach to data presentation facilitates interpretation and enhances the transparency of our results.

Secondly, the introduction lacks detailed explanations and citations regarding EEG-BF training. Adding more information and appropriate references about the basics and background of EEG-BF training would make the introduction more comprehensive and reader-friendly.

Thank you for your comment regarding the introduction. We have expanded this section to provide a more detailed discussion of the foundations and mechanisms of EEG biofeedback (EEG-BF) training, including its potential benefits for athletes. Additionally, we have incorporated relevant references to the literature to ensure a more comprehensive scientific context and improve readability for our audience. These changes aim to provide readers with a deeper understanding of the theoretical background of our study.

Once again, we thank you for your valuable feedback and constructive critique, which have helped us improve our manuscript. We believe that the implemented revisions have significantly enhanced the quality of the paper and meet the expectations of both the reviewer and potential readers.

Round 2

Reviewer 1 Report

Comments and Suggestions for Authors

The authors have replied my comments item-by-item. I have no more comments. Accept is my final decision